# EEG-Based Prediction of the Recovery of Carotid Blood Flow during Cardiopulmonary Resuscitation in a Swine Model

**DOI:** 10.3390/s21113650

**Published:** 2021-05-24

**Authors:** Heejin Kim, Ki Hong Kim, Ki Jeong Hong, Yunseo Ku, Sang Do Shin, Hee Chan Kim

**Affiliations:** 1Clinical Trials Center, Seoul National University Hospital, Seoul 03080, Korea; hjkim83@snu.ac.kr; 2Department of Emergency Medicine, Seoul National University Hospital, Seoul 03080, Korea; emphysiciankkh@gmail.com (K.H.K.); emkjhong@gmail.com (K.J.H.); sdshin@snu.ac.kr (S.D.S.); 3Department of Biomedical Engineering, Chungnam National University College of Medicine, 266 Munwha-ro, Jung-gu, Daejeon 35015, Korea; 4Department of Biomedical Engineering, Seoul National University College of Medicine, Seoul 03080, Korea; hckim@snu.ac.kr

**Keywords:** electroencephalogram (EEG), carotid blood flow (CBF), cardiopulmonary resuscitation (CPR), machine learning (ML), real-time feedback, classifier

## Abstract

The recovery of cerebral circulation during cardiopulmonary resuscitation (CPR) is important to improve the neurologic outcomes of cardiac arrest patients. To evaluate the feasibility of an electroencephalogram (EEG)-based prediction model as a CPR feedback indicator of high- or low-CBF carotid blood flow (CBF), the frontal EEG and hemodynamic data including CBF were measured during animal experiments with a ventricular fibrillation (VF) swine model. The most significant 10 EEG parameters in the time, frequency and entropy domains were determined by neighborhood component analysis and Student’s *t*-test for discriminating high- or low-CBF recovery with a division criterion of 30%. As a binary CBF classifier, the performances of logistic regression, support vector machine (SVM), k-nearest neighbor, random forest and multilayer perceptron algorithms were compared with eight-fold cross-validation. The three-order polynomial kernel-based SVM model showed the best accuracy of 0.853. The sensitivity, specificity, F1 score and area under the curve of the SVM model were 0.807, 0.906, 0.853 and 0.909, respectively. An automated CBF classifier derived from non-invasive EEG is feasible as a potential indicator of the CBF recovery during CPR in a VF swine model.

## 1. Introduction

More than 74 out-of-hospital arrest (OHCA) patients per 100,000 people are treated by emergency medical service in the United States every year [1]. Despite the improvement of the cardiopulmonary resuscitation (CPR) protocol and equipment for decades, only 8% of all OHCA patients survive with good neurological recovery. In practice, high-quality CPR is the key to enhancing OHCA patients’ clinical outcomes, including brain function recovery [2]. Multiple CPR feedback devices, which monitor compression depth, rate and chest recoil in real-time, have been used in the clinical field [3]. However, to evaluate the actual effectiveness of CPR, real-time monitoring of the patients’ physiologic data is highly recommended [4].

The American Heart Association (AHA) guidelines suggest three hemodynamic data, namely, end-tidal carbon dioxide (ETCO2), diastolic blood pressure (DBP) and coronary perfusion pressure (COPP), as outcome predictors for the quality of CPR [5,6]. Monitoring ETCO2 under endotracheal intubation is a widely used technique for CPR quality assessment [5]. However, improper placement of the endotracheal tube might obstruct the airway and lead to low ETCO2 levels [6]. DBP and COPP measurements are rarely possible in an OHCA setting due to the difficult surgical and catheterization procedures for central venous cannulation. Most of all, these hemodynamic data are mainly related to the recovery of the cardiac function or systemic circulation [6]. Maintaining sufficient blood flow to the brain is also crucial to recover the neurological function of OHCA patients. Otherwise, the resuscitated patients can suffer from neurological deficits [7]. Thus, monitoring techniques for blood supply to the brain, such as carotid blood flow (CBF), are necessary.

Several methodologies have been suggested to monitor cerebral circulation. Transcranial Doppler ultrasonography (TCD), which measures the velocity of cerebral blood flow, showed the potential to estimate the cerebral perfusion pressure (CEPP) [8,9] and was therefore applied to the resuscitated patients [10]. However, due to its bulky size and the difficulty of placing the ultrasonic probe, the TCD technique is not appropriate for the OHCA setting. Near-infrared spectroscopy (NIRS) technique also has been applied in OHCA setting [11]. Higher regional oxygen saturation (rSO2) during CPR could foretell ROSC, but does not necessarily mean brain function recovery [12]. As a simple and convenient alternative, the electroencephalogram (EEG) has been proposed as an outcome indicator for cerebral resuscitation and the recovery of brain activity. Non-invasive EEG could be easily measured in an ambulatory setting with portable EEG devices. In this manner, some EEG parameters, such as the bispectral index score, have been investigated for CPR quality assessment [13]. Table 1 introduces several state-of-the-art CPR feedback techniques. According to AHA consensus statement, CPR feedback techniques can be categorized into a performance-oriented and a patient-oriented perspective [14].

The former methods mainly observe how well CPR performance is performing [3], while the latter methods, such as capnography, TCD and NIRS, mainly observe the patient’s physiologic responses to CPR [15]. Each patient-oriented method can monitor the patient’s hemodynamic profile with its own strengths. However, those methods have difficulty in evaluating the resultant neurologic responses to determine neurological recovery or sequelae. In this respect, monitoring cerebral electrical activity by non-invasive EEG can be advantageous. In the previous study [17], the frontal EEG was measured without interrupting the operation of the CPR machine, and the relationship between the recovery rates of CBF and non-invasive EEG parameters before defibrillation attempts was investigated. We hypothesized that the CBF recovery might influence the brain cell activation, and such changes could be reflected on the EEG activity such as awakening or flattening [17,18]. Several time- and frequency-domain EEG parameters are positively correlated to the recovery rates of CBF, which is closely related to brain blood flow [19]. Based on these findings, this study aimed to evaluate the feasibility of machine learning (ML)-based estimation for high- and low- CBF recovery rates as a novel neurological CPR quality indicator. A supervised learning method and statistical analysis were performed to extract EEG parameters for the estimation model. Using these EEG parameters, five different ML models were developed with an oversampling technique to overcome the imbalance of training data and their classification performances were evaluated.

## 2. Materials and Methods

### 2.1. Ethical Statement

The animal test protocol was approved by the Institutional Animal Care and Use Committee of Seoul National University Hospital (IACUC number: 17-0106). All animal care complied with the Laboratory Animal Act of the Korean Ministry of Food and Drug Safety.

### 2.2. Study Design and Setting

A ventricular fibrillation (VF) swine model was designed to acquire the frontal EEG and hemodynamic data, including CBF [17]. This study assumed the scenario of a witnessed OHCA, a situation wherein cardiac arrest is recognized immediately and bystanders deliver manual chest compressions while waiting for the emergency medical team (EMT), who applies defibrillation shocks and administrates epinephrine, to arrive [20]. After VF was induced, the animals were left untreated for 1 min. Then, four consecutive 2-min chest compressions, by which the bystanders regarded as basic life support (BLS), were performed with the LUCAS2 Chest Compression System (Jolife AB, Lund, Sweden). At the end of the BLS sessions, a biphasic defibrillation shock of 200 J was applied to restart the heart. If a palpable pulse with organized QRS complexes appeared after the defibrillation, the monitoring session for 20 min started to confirm the sustained return of spontaneous circulation (ROSC) [12]. If a palpable pulse did not appear, or if VF occurred again during the monitoring session, one cycle of chest compression with a regular injection of 1 mg of epinephrine to increase the likelihood of ROSC once every 3 min [21], which is regarded as advanced cardiovascular life support (ACLS) performed by EMT, was done. After the single ACLS session, the defibrillation shock was also applied. A further ACLS cycle or monitoring session was executed according to the test protocol. Non-ROSC was confirmed if sustained ROSC was not achieved after the 10th ACLS session. The entire study design is shown in Figure 1.

### 2.3. Experimental Animal and Housing

Approximately 3 month-old domestic cross-bred female pigs (45.6 ± 2.4 kg) were studied (n = 8). The animals were maintained in a facility accredited by AAALAC International (#001169) in accordance with the Guide for the Care and Use of Laboratory Animals, 8th edition, NRC (2010) [22]. After a certified veterinarian adjudged them as healthy, the animals were made to fast overnight.

### 2.4. Surgical Preparation and Hemodynamic Measurements

The sedated animals were intubated and a carbon dioxide (CO2) analyzer (Capstar-100, CWE Inc., Ardmore, PA, USA) was installed to measure the continuous CO2 before the initialization of mechanical ventilation. An implantable perivascular probe (MA2PSB, Transonic Systems, Ithaca, NY, USA) and a single-channel perivascular flowmeter (T420, Transonic Systems, Ithaca, NY, USA) were utilized to measure the CBF in the ascending internal carotid artery. Pressure catheters (Mikro-tip pressure catheters, Millar, Houston, TX, USA) were inserted to measure the arterial blood pressure (ABP) and right arterial pressure. All hemodynamic data were gathered and saved in a data acquisition platform (PowerLab 16/35, ADInstruments, Dunedin, New Zealand).

A pace-making wire was inserted into the right ventricle to induce VF. Once VF occurred, mechanical ventilation was ceased and the animals were left without assistance for 1 min. Manual ventilation using a resuscitator bag (Ambu Resuscitators, Ambu A/S, Ballerup, Denmark) was performed at a rate of once per 6 *s*, a recommended ventilation rate for an intubated animal [23], while CPR and defibrillation attempts were being executed.

### 2.5. EEG Signal Acquisition

A lab-developed battery-powered single-channel EEG device with Ag/AgCl electrodes (MT-100, Kendall, Ontario, Canada) was installed on the forehead to measure the frontal EEG under the referential montage [17]. Reference and ground electrodes were attached on either side of the mastoid. An active electrode underneath the EEG device was affixed on the forehead. The raw EEG was bandpass-filtered for the 0.5−47 Hz frequency band and amplified with a gain of 12,000 *v*/*v*. The processed signal was digitized with the analogue-digital conversion process at a rate of 250 Hz and transmitted to a laptop via Bluetooth. The developed data acquisition software received and saved the EEG data simultaneously. The EEG during approximately 3-s-long pauses was segmented into three 2-s-long sub-epochs with 1.5-s overlaps to reduce variation; 0−2 s, 0.5−2.5 s and 1−3 s period.

### 2.6. Development of Machine-Learning Based Prediction Model

A data analysis and ML model development were performed using MATLAB (MATLAB R2020a, Mathworks, Natick, MA, USA). The recovery rates of CBF were determined with respect to the pre-arrest values (Equation (1)). The measured recovery rates of CBF were categorized into two groups: Group 0 with recovery rates less than 30%, and Group 1 with recovery rates equal to or greater than 30%. The recovery rate of 30% was set because 30−40% of normal cerebral blood flow could be the threshold for the recovery of the brain function and the flattering of the EEG [24]. The numbers of EEG epochs with CBF recovery groups are shown in Table 2. The epoch number of each experiment depends on when ROSC occurred in the test protocol (baseline, VF, 4 BLS and 10 ACLS). For example, ROSC occurred at the end of BLS sessions in animal 6 while animal 2 and 5 were not resuscitated until the 10th ACLS session, which led to a different number for each animal.
*Recovery rates (%) = (CBF during the specific CPR period/baseline CBF before VF)* * 100(1)

Initially, a total of 20 EEG parameters were prepared as candidate inputs for the prediction models. Three parameters (Magnitude, Rényi entropy, Log energy entropy), which showed a correlation coefficient of over 0.7 with the recovery rates of CBF in the previous study [17], were included. The final EEG parameters were selected in two stages. First, optimal candidate parameters for binary groups were determined through neighborhood component analysis (NCA) [25]. As a supervised learning method, NCA finds a linear transformation of input data and learns a distance metric. With such distance information, multivariate data are classified into a certain class over the data to maximize the classification performance. NCA can be utilized to select the most significant parameters of the EEG signals with feature ranking. Then, Student’s *t*-test was performed to find parameters that significantly differed between the two groups. The significance was set at *p* < 0.05.

The prediction models were established using three classical classifiers (logistic regression (LR), support vector machine (SVM) and k-nearest neighbors (KNN)), one graphical model (multilayer perceptron (MLP)) and one ensemble model (random forest (RF)), with EEG parameters as inputs and binary CBF groups as output. LR is a statistical model which uses a logistic function and estimates the probability of a certain class between binary variables [26]. For the LR model, the generalized logistic model using the binomial distribution was used. SVM is a supervised learning method that separates data with multi-dimensional hyperplanes to classify them into a certain class [27]. For the SVM model, the three-order polynomial kernel function with auto kernel scale and box constraint of 1 was applied. KNN is a non-parametric method that finds the number of nearest neighbors and classifies the data into the distinct class based on a similarity measure [28]. For the KNN model, the Euclidean method with the squared inversed weights was applied to measure distances between the data instances. A total of 9 neighboring data were investigated to classify each instance into one of the two CBF groups. RF uses multiple learning methods to achieve better prediction ability. It establishes a multitude of decision trees and classifies the data into a certain class by voting or averaging the outputs of the forests of decision trees [29]. An ensemble aggregation method with random subspace was used for the RF model. The number of ensemble learning cycle was 30. MLP is a feedforward artificial neural network composed of one or multiple layers of perceptron, a supervised learning algorithm of binary classification. It utilizes the backpropagation technique, a supervised learning method, for training and can distinguish data that are not linearly separable [30]. The MLP model with 10 hidden layers and scaled conjugated gradient backpropagation method was obtained. For all models, 5-fold cross validated classifiers were created. Before establishing the prediction models, bias problems due to imbalanced training data were resolved by augmenting the minority class with the synthetic minority oversampling technique (SMOTE), one of the most widely used oversampling techniques, which generates new synthetic data along the lines between randomly selected instances in the minority class [31]. Figure 2 shows the total developmental stages of the prediction models. The performance of the classifiers was evaluated with the leave-one-out of cross validation method. Each classifier model was established by merging data from seven of the eight animal data. The model performance was evaluated using the remaining data. The process was repeated eight times, and each classifier’s performance was averaged over eight animals.

### 2.7. Performance Evaluation

Confusion matrices were derived to evaluate the performance of the developed models. True positives (TP), true negatives (TN), false positives (FP) and false negatives (FN) were determined with the confusion matrix. The ground truth was established based on the actual CBF recovery rates and were categorized into the two groups, shown in Table 2. TP is the proportion of the known positive instances that are correctly predicted, and TN is the proportion of the known negative instances that are correctly predicted. FP and FN represent the proportion of the incorrect positive and negative predictions, respectively. As primary evaluation metrics [32], the accuracy, sensitivity and specificity were calculated as follows.
*Accuracy = (TP + TN)/(TP + TN + FP + FN)*(2)
*Sensitivity = TP/(TP + FN)*(3)
*Specificity = TN/(TN + FP)*(4)

For the second metric, the F1 score [33], the harmonic mean of precision (TP/(TP + FP)) and recall (TP/(TP + FN), also known as sensitivity) is derived as follows.
*F1 score = 2 * (precision * recall)/(precision + recall)*(5)

Additionally, a receiver operating characteristic (ROC) curve and area under the curve (AUC) were obtained with SPSS (IBM SPSS Statistics 25, IBM SPSS Statistics, New York, NY, USA) to evaluate the classification ability of the models while the discrimination threshold was varied. The ROC curve is created by plotting the true positive rate ((TP/(TP + FN), also known as sensitivity) against the false positive rate ((FP/(FP + TN), also known as (1—specificity)) at different threshold levels. The AUC is defined as the area under the ROC curve.

## 3. Results

### 3.1. EEG Changes throughout the Experiments

Distinctive EEG changes were observed in the test protocol. Time-domain EEG waveforms and frequency responses measured from animal 1 are shown in Figure 3. Before VF, the amplitude of irregular EEG activity exceeded ±20 μV (Figure 3a). Once VF occurred, the amplitude dropped in approximately 15 *s* and became less than <±5 μV at the end of untreated VF (Figure 3b). During CPR sessions, periodic motion artifact exceeding ±50 μV were observed (Figure 3c). Frequency components associated with the rate of CPR dominated the power spectrum (Figure 3g). However, EEG activity with higher amplitude and increased frequency components was observed after the 1st ACLS session (Figure 3d,h).

### 3.2. EEG Parameters for Modeling

A total of 10 EEG parameters were selected through NCA and Student’s *t*-test. The feature ranking was determined after the NCA, but parameters with *p* over 0.05 were discarded after the Student’s *t*-test regardless of the ranking. The top 10 significant parameters were finally adopted for the modeling process. Table 3 summarizes the feature selection process. In total, a time-domain parameter (Magnitude), six frequency-domain parameters (DeltaR, DAR, DTABR, DeltaPR, BcSEF and BG_Alpha+) and three entropy indices (Spectral entropy, Rényi entropy and Log energy entropy) were used in the modeling process. It is noteworthy that three parameters (Magnitude, Rényi entropy and Log energy entropy) consistently showed a correlation coefficient greater than 0.7 with the recovery rates of CBF in the previous study [17]. The parameters’ information, including definitions, medians and interquartile ranges between groups, are shown in Table 4.

### 3.3. Performances of Prediction Models

Prediction models were derived based on the different ML and neural network algorithms. During the training time, SMOTE was applied to overcome the data imbalance issue. The result of the SMOTE application is shown in Figure 4.

Of the 10 EEG parameters, Log energy entropy and DeltaPR are presented on a scatterplot as examples. In the original dataset, the number of majority class, group 0 (blue circle markers, n = 117), is approximately twice the number of minority class, group 1 (red circle markers, n = 71). After SMOTE was applied, newly augmented data instances (black diamond markers, n = 71), were introduced to the minority class (group 1). The sample size of group 0 and group 1 became comparable (n = 117 vs. n = 114) and prediction models were established with the augmented dataset. Figure 5 and Figure 6 present the confusion matrices and the ROC curve of the prediction models, respectively. The overall performance is summarized with six evaluation indices in Table 5. The accuracy ranged from 0.813 to 0.853, while the sensitivity ranged from 0.689 to 0.807 and the specificity ranged from 0.877 to 0.953. The precision ranged from 0.880 to 0.943 and the F1 score ranged from 0.796 to 0.853. In terms of accuracy, the 3-order polynomial kernel-based SVM model showed the best performance. Furthermore, all five classifiers showed high AUC values of over 0.9.

## 4. Discussion

### 4.1. EEG Parameters and Classification Model of Performances

In this study, ML-based prediction models of CBF recovery using non-invasive EEG parameters were investigated as potential indicators of the recovery of CBF during CPR in a VF swine model. Certain levels of cerebral circulation should be achieved to maintain the metabolism of brain cells and to expect recovery of consciousness and brain function [24]. Currently, OHCA patients are routinely treated according to the guidelines, mainly focused on the recovery of the cardiopulmonary function [5,6]. In such a situation, monitoring systemic circulation might be possible with continuous ETCO2 measurement. The ETCO2 levels are greatly affected by changes in cardiac output (CO) under the prolonged low-flow condition. However, ETCO2 might not reflect cerebral blood flow because CO does not correlate with cerebral blood flow [16]. Along with the conventional ETCO2 measurement, the requirement of new techniques for monitoring cerebral circulation has been suggested [18].

Several non-invasive methodologies have been studied to measure cerebral hemodynamics. Especially, TCD methods have been reported to estimate CEPP [8]. Despite competitive performance, however, these techniques might not be applicable to the out-of-hospital CPR setting due to the difficulty in finding the vessels and placing the probe correctly on patient’s head, as well as due to the invasive procedure for ABP measurement. NIRS techniques have been used to measure the quality of CPR by measuring rSO2, which can be an indicator of the occurrence of ROSC [11]. Even though rSO2 can directly show the cerebral oxygenation levels, it can be distorted from extracranial blood flow [34] and might represent systemic circulation rather than cerebral circulation during CPR [35]. In addition, the relationship between higher rSO2 levels and favorable neurological outcomes remains unclear [11]. In this respect, non-invasive EEG can be an alternative. Basically, the EEG signal represents the functional dynamics of the brain cells. Different levels of amplitude, distribution of frequency components and entropy indices of the EEG are affected by the blood flow throughout the brain [36,37]. Moreover, a portable EEG device is small and lightweight, which enables EMTs to easily bring and apply the device to OHCA patients. The automated prediction model of the CBF recovery is beneficial because it does not require skillful operation or additional measurements. However, the EEG signal is prone to external artifact resulting from the mechanical chest compressions during CPR. To avoid such unfavorable effect, the prediction model only exploited an artifact-free EEG signal measured during approximately 3-s-long pauses before defibrillation attempts [17]. All the selected EEG parameters significantly differed between the two groups. For the time-domain parameters, the Magnitude showed bigger amplitudes in group 1, which indicates that more wakeful EEG patterns appeared under the improved blood flow [13]. For the frequency-domain parameters, the higher frequency components over 8 Hz increased, while the lower frequency components decreased in the high-CBF group [17]. DeltaR, DAR and DTABR, which compared the lower and higher frequency bands, consistently reflected such trend. Furthermore, DeltaPR decreased in the high-CBF group. These four parameters contain the delta components (1−4 Hz).

It is known that the power decrease in the delta band is related to the conscious recovery [38]. Such a change can be utilized in cerebral circulation assessment during CPR. Simultaneously, BcSEF consistently showed higher values in the high-CBF group. The spectral edge frequency (SEF), a classical parameter that refers to the frequency below which 95% of the total spectral power is occupied, has been used to measure the depth of anesthesia and sedation [39]. As the higher frequency components increased, the power spectrum was re-distributed toward higher frequencies, which led to an increased SEF [39]. In this study, the burst suppression ratio (BSR)-compensated SEF (BcSEF), defined as (1—BSR/100), was applied instead of an original SEF. BSR was excluded due to its low feature ranking, but, it showed the highest value of 100 mostly in low-CBF group, which induced a zero-level BcSEF. Thus, a stark difference was observed in this parameter between the two groups. The increased spectral complexity levels were also found with three entropy indices in the high-CBF group. By exploiting these differences in the EEG parameters between the two CBF groups, the proposed prediction models achieved AUC values of at least 0.909, which indicates a high classification performance [40]. In this study, three types of classifier models, including classic classifiers (LR, SVM and KNN), a graphical classifier (MLP) and an ensemble classifier (RF), were implemented and compared. The SVM model based on a polynomial kernel function showed the best result, but the other models also showed competitive performance with an AUC of over 0.9.

### 4.2. Clinical Implication and Limitations

Currently, several evaluation scoring systems, such as the cerebral performance category (CPC), are used during post-resuscitation treatment [41]. To our knowledge, there are no reports of feedback indicators for cerebral recovery during CPR. When a low CBF level is diagnosed in OHCA patients, EMTs could alter CPR strategies to increase blood supply to the brain, which makes early recovery of the brain function possible, for example, by injecting epinephrine [21] or guiding the Trendelenburg position [42]. If the EEG change during CPR is closely related to CBF recovery, the neurological outcomes of patients with cardiac arrest can be estimated noninvasively during CPR. Follow-up studies involving real patients should be warranted. The proposed EEG parameters and ML model development process such as an oversampling method or hyperparameters can be a good starting point for the follow-up clinical studies. Further studies can include a regression approach to estimate quantitative CBF recovery rates with a sufficient number of patients.

This study had several limitations. First, the prediction model was established and evaluated with a limited size of eight animals. The proposed prediction model should be regarded as preliminary, and its performance should be evaluated after further human clinical trials. Second, the VF duration of 1 min is short because this study assumed a witnessed OHCA setting. If the untreated VF duration becomes longer, minimal or no recovery of the EEG activity could be achieved. The EEG parameters obtained from the various VF durations should be scrutinized to establish a more practical classifier model.

## 5. Conclusions

In an experimental VF swine model, ML-based prediction models of CBF recovery using EEG parameters were developed to estimate the status of cerebral circulation during CPR. The proposed models classified the binary groups of the recovery of CBF with an accuracy of 0.834, a sensitivity of 0.765, a specificity of 0.911 and an AUC of 0.923, on average the results showed that non-invasive EEG-based automated screening of CBF recovery has the potential for real-time feedback of cerebral circulation during CPR.

## Figures and Tables

**Figure 1 sensors-21-03650-f001:**
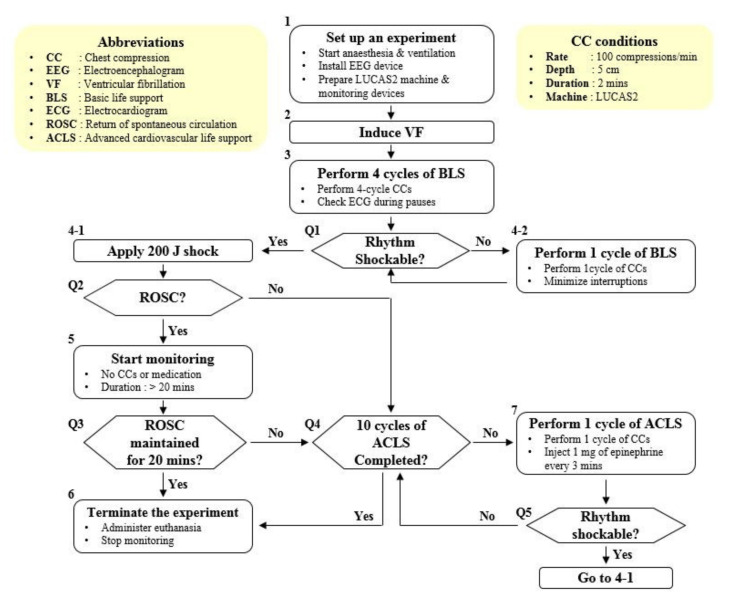
Flow chart of the study design [17]: After the initial experimental setup was completed (Step 1), VF was induced and an animal was left untreated for 1 min (Step 2). Four consecutive 2−minute BLS sessions were performed by the bystanders (Step 3). The EMT was supposed to arrive at the site and ECG was checked (Step Q1). If ECG was shockable, a biphasic defibrillation shock of 200 J was applied by the EMT (Step 4-1). Otherwise, an additional BLS session was performed (Step 4-2) and ECG was checked again (Step Q1). If ROSC occurred after the defibrillation attempt (Step Q2), the monitoring session was initiated (Step 5) and sustained ROSC was confirmed if ROSC lasted for 20 min (Step Q3 → Step 6). However, if ROSC did not occurred (Step Q2) or VF occurred again during the monitoring session (Step Q3), ACLS sessions for 2 min with an injection of epinephrine were performed up to 10 times (Step Q4, 7). If ROSC occurred (Step Q2), the monitoring session was initiated (Step 5). Otherwise, an additional ACLS session was performed (Step Q4, 7). Non-ROSC was confirmed if sustained ROSC was not achieved after the 10th ACLS session (Step Q4 → Step 6).

**Figure 2 sensors-21-03650-f002:**
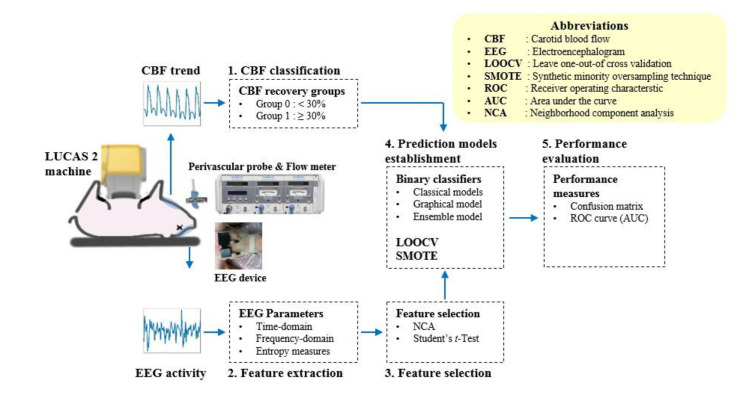
Total developmental stages of the prediction models; from animal experiments to performance evaluation: the frontal EEG and CBF data were acquired during the process of CPR. CBF recovery rates were calculated and categorized into two groups with a division criterion of 30%. Time- and frequency-domain EEG parameters and entropy endices were determined. The top 10 significant EEG parameters were selected through the NCA and Student’s *t*-Test. An EEG-based binary classifier for CBF recovery was established based on LOOCV and SMOTE methods. Finally, the performances of prediction models were evaluated with the confusion matrix and ROC curve.

**Figure 3 sensors-21-03650-f003:**
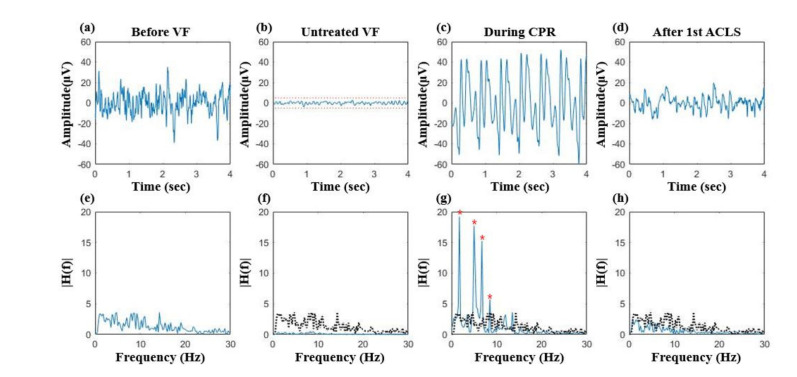
EEG changes in animal 1: (**a**) irregular baseline EEG activity before VF, (**b**) isoelectric EEG pattern during untreated VF, red dotted lines denote the limits of the isoelectric state (±5 μV), (**c**) severely contaminated signal during CPR, (**d**) recovered EEG activity after the 1st ACLS, (**e**) frequency response of the baseline EEG before VF, (**f**) frequency response of the isoelectric EEG during untreated VF, (**g**) frequency response of CPR artifacts, red marks denote the frequency components of CPR artifacts, (**h**) frequency response of the EEG after the 1st ACLS, a black dotted line in (**f**−**h**) is the frequency responses of the baseline EEG (**e**).

**Figure 4 sensors-21-03650-f004:**
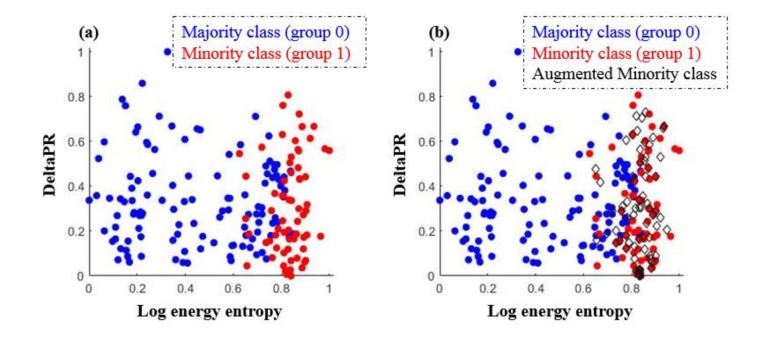
An oversampling example by SMOTE application: (**a**) Before applying SMOTE, (**b**) After applying SMOTE.

**Figure 5 sensors-21-03650-f005:**
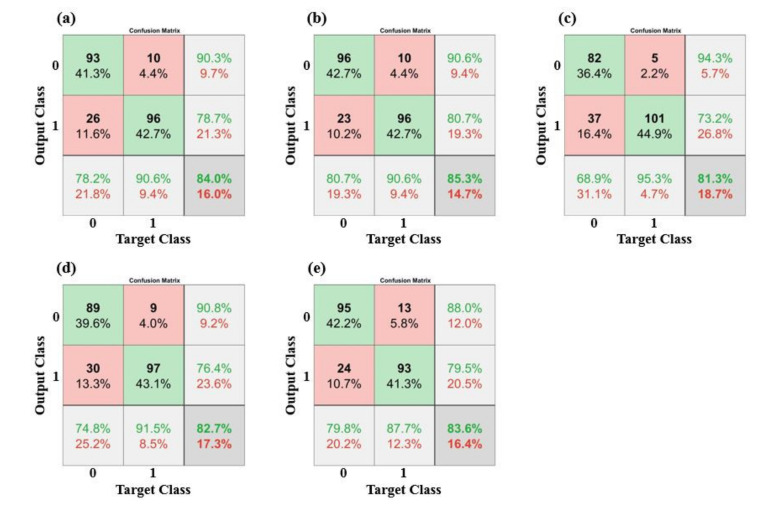
Confusion matrices of classifier models: (**a**) LR, (**b**) SVM, (**c**) KNN, (**d**) RF, (**e**) MLP.

**Figure 6 sensors-21-03650-f006:**
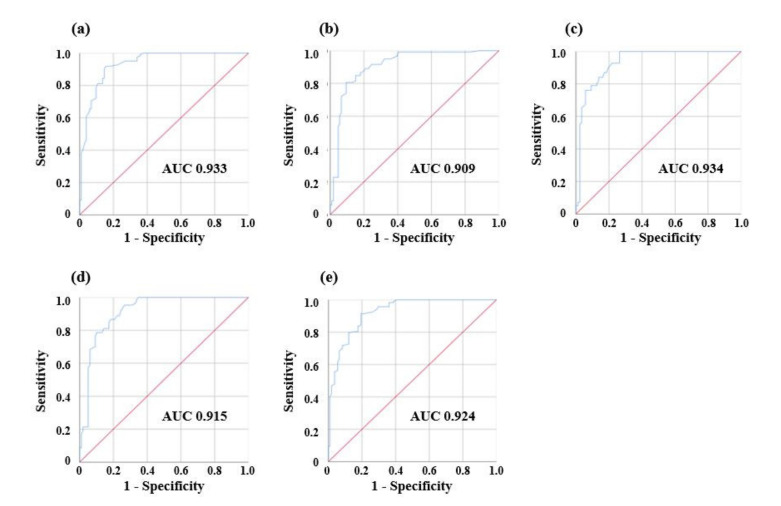
The ROC curves and AUC values of classifier models: (**a**) LR, (**b**) SVM, (**c**) KNN, (**d**) RF, (**e**) MLP.

**Table 1 sensors-21-03650-t001:** Various state-of-the-art CPR feedback techniques.

Category	Method	Measurement	Characteristics
Performance-oriented	Accelero-meter	Compression depth, rate	- Can guide exact compression depths and rates with multiple audiovisual feedback signals [3]. - Multiple devices are used, currently. - Cannot reveal physiologic responses of the patients [15].
	Force sensor	Chest recoil	
Patient-oriented	Capno-graphy	ETCO2	- Can reveal systemic circulation through endotracheal intubation [5] - Cannot reveal cerebral blood flow exactly [16].
	TCD	Velocity of CBF	- Can evaluate cerebral blood flow quantitatively [9] - Might require an additional measurement and be inappropriate in OHCA setting [8].
	NIRS	rSO2	- Higher rSO2 could indicate ROSC [11] - Favorable neurological outcomes are not guaranteed [12].
	EEG	Cerebral electrical activity	- Can reveal brain function recovery. - Can be easily contaminated due to external artifacts [13].

Abbreviation: ETCO2, end-tidal CO2; TCD, transcranial Doppler; OHCA, out-of-hospital cardiac arrest; NIRS, near-infrared spectroscopy; CBF, cerebral blood flow; rSO2, regional oxygen saturation; ROSC, return of spontaneous circulation; EEG, electroencephalogram.

**Table 2 sensors-21-03650-t002:** Number of EEG epochs with CBF recovery groups.

CBF Recovery Degree	Class	Amount of Acquired Data in Each Experiment									
		Test Number	1	2	3	4	5	6	7	8	Total
Lower (<30%)	group 0		0	40	0	0	44	0	2	33	119
Higher (≥30%)	group 1		15	1	28	12	1	14	35	0	106
	**All**	**Total amount**	15	41	28	12	45	14	37	33	225

Abbreviation: CBF, carotid blood flow.

**Table 3 sensors-21-03650-t003:** The result of feature selection process among 20 EEG parameters.

EEG Parameters	Definition	NCA Feature Ranking	*p-*Value	Result of Selection
BSR	Burst suppression ratio	18	<0.001	Excluded
Magnitude	Maximal amplitude during the epoch(unit: µV)	11	<0.001	Selected
SynchFastSlow	Relative synchrony of fast and slow wave	17	0.264	Excluded
BetaR	log(P_20–47 Hz_/P_11–20 Hz_)	4	0.864	Excluded after Student’s *t*-test
DeltaR	log(P_8–20 Hz_/P_1–4 Hz_)	5	<0.001	Selected
DAR	log(P_1–4 Hz_/P_8–13 Hz_)	7	<0.001	Selected
DTABR	log(P_1–8 Hz_/P_8–30 Hz_)	6	<0.001	Selected
BcSEF	Burst suppression ratio-compensated spectral edge frequency 95	1	<0.001	Selected
ExtraPR	P_40–47 Hz_/P_1–47 Hz_	19	0.675	Excluded
AlphaPR	P_8–13 Hz_/P_1–47 Hz_	12	0.439	Excluded
BetaPR	P_13–30 Hz_/P_1–47 Hz_	16	<0.001	Excluded
DeltaPR	P_1–4 Hz_/P_1–47 Hz_	3	0.014	Selected
ThetaPR	P_4–8 Hz_/P_1–47 Hz_	15	<0.001	Excluded
GammaPR	P_30–47 Hz_/P_1–47 Hz_	13	0.001	Excluded
Shannon entropy	$\left(-1\right)*\overset{n}{\underset{k=0}{\sum }}\left(p\left(x_{i}\right)*\log\left(p\left(x_{i}\right)\right)\right)$	14	0.607	Excluded
Log energy entropy	$\overset{n}{\underset{i=1}{\sum }}\log{\left(p\left(x_{i}\right)\right)}^{2}\;$	10	<0.001	Selected
Spectral entropy	$\overset{n}{\underset{i=1}{\sum }}p\left(x_{i}\right)\log\left(1/p\left(x_{i}\right)\right)$	2	0.013	Selected
Rényi entropy	$1/\left(1-\alpha \right)*\log(\overset{n}{\underset{i=1}{\sum }}p{\left(x_{i}\right)}^{\alpha }),\left(\alpha =0.5\right)$	8	<0.001	Selected
Ratio05	Percentage of data whose amplitude is under ±5 µV	20	<0.001	Excluded
BG_Alpha+	P_8–47 Hz_/P_1–47 Hz_	9	<0.001	Selected

Abbreviation: NCA, neighborhood component analysis; P_a-b Hz_, the sum of spectral power from a−b Hz; $p\left(x_{i}\right)$
, probability distribution function of signal $x_{i}$, probability distribution function of signal $x_{i}$.

**Table 4 sensors-21-03650-t004:** Final 10 EEG parameters used to establish the prediction models; definition, median, interquartile range and *p-*value.

EEG Parameters	Definition	Median (IQR)		*p-*Value
		Group 0	Group 1	
Magnitude	Maximal amplitude during the epoch (unit: µV)	4.08 (2.52 14.28)	18.55 (15.72 22.29)	<0.001
DeltaR	log(P_8–20 Hz_/P_1–4 Hz_)	0.00 (−0.30 0.31)	0.25 (−0.08 0.72)	<0.001
DAR	log(P_1–4 Hz_/P_8–13 Hz_)	0.20 (−0.10 0.49)	−0.07 (−0.42 0.36)	<0.001
DeltaPR	log(P_1–4 Hz_/P_1–47 Hz_)	0.28 (0.17 0.41)	0.20 (0.08 0.41)	0.014
DTABR	log(P_1–8 Hz_/P_8–30 Hz_)	0.18 (−0.10 0.45)	−0.06 (−0.31 0.21)	<0.001
BcSEF	Burst suppression ratio-compensated spectral edge frequency 95	0.00 (0.00 19.50)	27.0 (22.0 30.0)	<0.001
BG_Alpha+	P_8–47 Hz_/P_1–47 Hz_	39.90 (26.54 55.94)	54.53 (39.07 67.68)	<0.001
Spectral entropy	$\overset{n}{\underset{i=1}{\sum }}p\left(x_{i}\right)\log\left(1/p\left(x_{i}\right)\right)$	0.78 (0.76 0.80)	0.79 (0.77 0.81)	0.013
Rényientropy	$1/\left(1-\alpha \right)*\log(\overset{n}{\underset{i=1}{\sum }}p{\left(x_{i}\right)}^{\alpha }),\;\left(\alpha =0.5\right)$	6.93 (5.57 9.06)	9.83 (9.20 10.11)	<0.001
Log energyentropy	$\overset{n}{\underset{i=1}{\sum }}\log{\left(p\left(x_{i}\right)\right)}^{2}\;$	−241.24 (−1014.8 729.0)	1122.30 (836.8 1304.1)	<0.001

Abbreviation: IQR, interquartile range; P_a–b Hz_, the sum of spectral power from a−b Hz; $p\left(x_{i}\right)$

**Table 5 sensors-21-03650-t005:** Performances of prediction models with the different algorithms.

Performance	LR	SVM	KNN	RF	MLP
Accuracy	0.840	0.853	0.813	0.827	0.836
Sensitivity	0.782	0.807	0.689	0.748	0.798
Specificity	0.906	0.906	0.953	0.915	0.877
Precision	0.903	0.906	0.943	0.908	0.880
F1 score	0.838	0.853	0.796	0.820	0.837
AUC	0.933	0.909	0.934	0.915	0.924

Abbreviation: LR, logistic regression; SVM, support vector machine; KNN, k-nearest neighbor; RF, random forest; AUC, area under the curve; MLP, multilayer perceptron.

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
