# Peer review of "EEG-Based Prediction of the Recovery of Carotid Blood Flow during Cardiopulmonary Resuscitation in a Swine Model"

_sensors, 2021, doi:10.3390/s21113650_

Round 1
Reviewer 1 Report
The manuscript has been substantially improved; many details were included to improve readability.
In general, the author made a great effort to address all my remarks.
The paper is technically sound and deserves acceptance.
Author Response
Thank you very much for giving us the opportunity to strengthen our manuscript with your valuable comments.
Reviewer 2 Report
The authors present a comparison of five well-known machine learning-based methods applied to predict carotid blood flow recovery during cardiopulmonary resuscitation. For this, electroencephalogram parameters are used. The methodology followed seems appropriate. However, an overview of the state of the art on which to make a comparison/discussion is not provided. Table 1 presents methods in general without referencing similar works with their strengths and weaknesses that have motivated carrying out this work. This brief review will allow highlighting the findings and novelty of this work. The conclusions obtained are also very limited. They must be completed with information about the potential clinical implications of the findings. As minor comments, the abstract should be a concise summary without mentioning weaknesses, future work, or conclusions (lines 25 to 28).
Author Response

(The authors gave the same response as above.)

Reviewer 3 Report
Ok about your responses to my review. I doubt that your study from a practical point of view will be of great use.
Author Response
Thank you very much for giving us the opportunity to strengthen our manuscript with your valuable comments. We will conduct further clinical research to realize the proposed methods in the real world. We sincerely appreciate your comments to strengthen the logic of our manuscript.
Round 2
Reviewer 2 Report
The changes made are really simple and do not meet my expectations in terms of improving an already poor work, which is reflected in its poor conclusions. As this is the policy of the journal in terms of deadlines and quality, OK.
This manuscript is a resubmission of an earlier submission. The following is a list of the peer review reports and author responses from that submission.
Round 1
Reviewer 1 Report
sensors-1176144
EEG-based Prediction of the Recovery of Carotid Blood Flow during Cardiopulmonary Resuscitation in a Swine Model
I looked the paper and misunderstand which work the authors done. Some specific comments listed as follows should be considered and tackled firstly by the authors.
(1) From the Title, it looks that might some merits of the manuscript. However, too many cited reference throughout the manuscript. Therefore, many works have been done by others. What’s the novelty of your manuscript?
(2) Abstract not clearly show the reader which work you done!
(3) Too many abbreviations also show the reader that this is not a novel manuscript but the combination of many methods!
Based on the above analysis, I strongly recommend the manuscript must be rejected.
Reviewer 2 Report
In this work, a binary classification is performed to assess the quality of cardiopulmonary resuscitation (CPR). In general, the content of this article could be better explained.
I have some questions:
1) You have commented in the introduction that Doppler ultrasound is not adequate because of its size, etc. If I'm not mistaken, that technique could offer quantitative control. In theory, your EEG-based method could also offer quantitative control. However, you have decided to implement a binary classification. Why didn't you implement a quantitative control? Why not implement a regression instead of a classification?
2) Is it possible to visualize any change in the EEG signal? If so, could you provide a figure in which a change in the EEG can be observed? For example, normal EEG versus CPR EEG
3) Could you explain the concept of "recovery rate"? Could you provide a formula? Could you explain table 1? Mainly, the concept of "number of cases".
4) Could you describe the hyperparameters selected for the different classification methods? How many examples per animal have you used? If the number of examples is small, your methods may not be well trained.
5) You could have used Random Forest to select significant variables. Do you use a sliding window to determine the variables? If so, explain what that sliding window looks like.
6) With your algorithm, you could diagnose a low level of CPR. In this case, as you say in the discussion, the doctor could use epinephrine or the Trendelenburg position. Why not use those solutions at the beginning?
Minor:
A) Paragraph in "Introduction": "In our previous study, we also measured the frontal EEG without interrupting the operation of the CPR machine and investigated the relationship between the recovery rates of CBF and non-invasive EEG parameters before defibrillation attempts 23". I think you could move the number 23 to the beginning: 'In our previous study [23],'
B) I think you could write your article following the mdpi rules. In this case it is easier to review the article because the number of each line appears.
Reviewer 3 Report
Dear authors,
I read your manuscript with great interest. Additional sources of potential prediction tools during and after cardiac arrest are urgently needed.
The manuscript is well-written and appears scientifically sound. I have a few points that need addressing, please see below. However, I can only judge the manuscript in terms of clinical applications and the cardiac arrest surroundings - for detailed comments on the theoretical part on the prediction model development, a respective expert should be consulted.
-) Introduction: When you talk about current recommendations and guidelines, please do not cite out-dated ons, but rather use the current recommendations by ILCOR, AHA or ERC.
-) Introduction: "[...] the most widely used technique for CPR Quality assessment" This needs to be cited.
-) Introduction: "[...] been suggested to monitor the cerebral circulation" I would delete "the".
-) Methods: Was there a case when amiodarone should have been given?
-) Methods: How did you come up with the numer of 8 assessed animals? Did you perform a power calculation?
-) Methods: "[...] a rate of once per 10 compressions [...]" Why? Why not 30:2? Please clarify.
-) Table 1: Please explain the table in greater detail, it is not intuitive.
-) All tables: Please provide more comprehensive legends, and expalin all abbreviations.
-) Discussion: The discussion could use subheadings to structure it more.
-) Discussion: Please provide up-to-date information on NIRS. E.g., citations 58 and 59 are rather old, for example refer to https://pubmed.ncbi.nlm.nih.gov/29410191/
-) Discussion: At the end, more information on the potential clinical implications of your findings would be warranted. (This also applies to the Conclusion).
Reviewer 4 Report
Include de contributions of this manuscript at the end of the introduction, as well as the weakness of the proposed method.
It is recommended to include a Table with most representative state-of-the-art methods related to this study. This review will allow highlighting the findings of this study.
Figure 1 must include a more descriptive caption; the labels next to the blocks should be mentioned.
Please add a representative picture from an EEG.
For the Performance evaluation, how was built the ground truth?
The results are too synthetically presented; it is recommended to include a more complete description or new results if they are available.
In the Tables, please highlight or put in boldface the most representative values.
Reviewer 5 Report
In this work, five machine learning-based prediction models of carotid blood flow recovery using electroencephalogram parameters were investigated as potential indicators of the recovery of carotid blood flow during cardiopulmonary resuscitation in a ventricular fibrillation swine model. The objective of the work is not completely. If the objective is only to compare five well-known classification methods based on machine learning, it does not contribute anything except a simple comparison between these methods. An overview of the state of the art on which to make a comparison/discussion is not provided. The conclusions obtained with this work are also very limited and based on a small number of tests which gives it little relevance. Thus, I do not see any interest to potential readers.
As minor concerns:
- Figure 1 has already been previously published by the authors.
- The title of Figure 2 should be more descriptive.
- The use of first persons (i.e., “we”, “their”, possessives, and so on) should be avoided and can preferably be expressed by the passive voice or other ways.
- I would recommend sending the work to a more clinical journal in the medical field.